# Novel Coatings to Minimize Corrosion of Titanium in Oral Biofilm

**DOI:** 10.3390/ma14020342

**Published:** 2021-01-12

**Authors:** Samira Esteves Afonso Camargo, Tanaya Roy, Xinyi Xia, Chaker Fares, Shu-Min Hsu, Fan Ren, Arthur E. Clark, Dan Neal, Josephine F. Esquivel-Upshaw

**Affiliations:** 1Department of Restorative Dental Sciences, Division of Prosthodontics, University of Florida College of Dentistry, Gainesville, FL 32610, USA; scamargo@dental.ufl.edu (S.E.A.C.); shuminhsu@ufl.edu (S.-M.H.); BCLARK@dental.ufl.edu (A.E.C.); 2Department of Materials Science Engineering, Herbert Wertheim College of Engineering, University of Florida, Gainesville, FL 32611, USA; t.roy@ufl.edu; 3Department of Chemical Engineering, Herbert Wertheim College of Engineering, University of Florida, Gainesville, FL 32611, USA; xiaxinyi@ufl.edu (X.X.); c.fares@ufl.edu (C.F.); fren@che.ufl.edu (F.R.); 4Department of Neurosurgery, University of Florida College of Medicine, Gainesville, FL 32610, USA; dneal@ufl.edu

**Keywords:** corrosion, implant, coating, SiC, TiN

## Abstract

The aim of this work is to investigate the effects produced by polymicrobial biofilm (*Porphyromonas gingivalis*, *Streptococcus mutans*, *Streptococcus sanguinis*, and *Streptococcus salivarius*) on the corrosion behavior of titanium dental implants. Pure titanium disks were polished and coated with titanium nitride (TiN) and silicon carbide (SiC) along with their quarternized versions. Next, the disks were cultivated in culture medium (BHI) with *P. gingivalis*, *S. mutans*, *S. sanguinis*, and *S. salivarius* and incubated anaerobically at 37 °C for 30 days. Titanium corrosion was evaluated through surface observation using Scanning Electron Microscope (SEM) and Atomic Force Microscopy (AFM). Furthermore, the Ti release in the medium was evaluated by Inductively Coupled Plasma Mass Spectrometry (ICP-MS). SEM images showed that coated Ti disks exhibited lower corrosion compared to non-coated disks, except for the quartenized TiN. This was confirmed by AFM, where the roughness was higher in non-coated Ti disks. ICP showed that Ti levels were low in all coating disks. These results indicate that these SiC and TiN-based coatings could be a useful tool to reduce surface corrosion on titanium implant surfaces.

## 1. Introduction

Dental implants can be susceptible to peri-implantitis and peri-mucositis, which are diseases of the supporting structures of the implant. That result in bone loss and tissue inflammation. The prevalence of peri-implantitis was shown to be as high as 85.0% and the incidence at 43.9% within five years [1]. The most effective treatment for peri-implantitis is prevention by minimizing bacterial colonization and implementing implant maintenance regimens. Therefore, there is a critical need to develop preventive strategies, such as implant surface modification, to minimize bacterial colonization, proliferation, and surface corrosion of the implant.

The degradation of titanium (Ti) and the corresponding native surface oxide has been shown to result in the release of Ti ions into the surrounding tissue, which can initiate inflammatory reactions and often lead to peri-implantitis. To further support this claim, a previous study found that implants removed from a patient’s oral cavity, due to peri-implantitis, often demonstrated pitting, a resulting from titanium corrosion [2,3]. Once corrosion causes a material loss, in vitro investigations have observed Ti corrosion by Scanning Electron Microscopy (SEM) [4,5,6,7], energy dispersive spectroscopy (EDS) [5,6], X-ray diffraction (XRD) [3,4,5], X-ray photoelectron spectroscopy (XPS) [5,7], and Atomic Force Microscopy (AFM) [6]. Furthermore, Inductively Coupled Plasma Mass Spectrometry (ICP-MS) can measure the Ti levels release in solution from a corrosion process [3].

Previous studies have shown that oral bacteria not only contribute to titanium corrosion but can also expedite the process [3,4]. *Porphyromonas gingivalis* (*P. gingivalis*), amongst some other microorganisms, is present in high numbers during biofilm organization on titanium implants and have been shown to be a part of the characteristic bacterial profile in peri-implantitis [8,9,10,11]. Oral bacteria such as *Actinomyces naeslundii* (*A. naeslundii*) and *Streptococcus mutans* (*S. mutans*) decrease the pH of their environment by generating organic acids, which break down the oxide layer and compromise the corrosion resistance of titanium [7,12].

The microorganisms have several virulent agents and release volatile sulfur compounds, such as methyl mercaptan, hydrogen sulfide, and dimethyl sulfide, resulting from their metabolism [13,14]. These virulent agents can influence titanium corrosion, depending on their concentration and pH [3]. Titanium implants can have mechanical and biological complications, due to corrosion [15,16]. Some studies have noticed that the association of corrosion, mechanical stress, and bacteria can initiate peri-implantitis and contribute to implant failure [3,4,15].

Corrosion fatigue fracture is another outcome of corrosion to the implant. Bacterial corrosion can produce weak points that progress to fatigue cracks, which can propagate with chewing cycles. When the crack extends sufficiently, the implant material could fail to withstand the load, resulting in implant fractures [17,18].

Dental titanium implants coated with titanium nitride (TiN) and silicon carbide (SiC) along with their quarternized versions can be used to decrease surface corrosion, improve fracture resistance, and minimize bacterial adhesion [5,6].

Therefore, the purpose of this research is to determine the influence of *P. gingivalis*, *S. mutans*, *S. sanguinis*, and *S. salivarius* on the corrosion behavior of titanium and whether different coatings on titanium can minimize surface corrosion.

## 2. Materials and Methods

### 2.1. Titanium Sample Preparation

Titanium rods at high purity (0.9999, TMS Titanium, Poway, CA, USA) were cut with Buehler Isomet 2000 into disks and polished to a grit size of 600 to 11 mm × 2 mm. First, these disks were cleaned with acetone in an ultrasonic bath, then rinsed with isopropyl alcohol, and dried with compressed nitrogen. The disks were then treated with ozone to remove any surface carbon contamination.

### 2.2. Coating Process

Forty-eight pre-cleaned titanium disks were coated with either TiN, quarternized titanium nitride coated titanium (QTiN), silicon carbide (SiC), or quaternized SiC (QSiC) (4 disks per group). The remaining 12 disks were left as controls.

To coat samples with 200 nm of SiC, a plasma-enhanced chemical vapor deposition system (PECVD, PlasmaTherm 790, Saint Petersburg, FL, USA) was utilized. The gas precursors used for SiC deposition were Methane (CH_4_) and Silane (SiH_4_) at a deposition temperature of 300 °C. Prior to quaternizing the SiC surface, the SiC was exposed to a 400 vW ammonia plasma for 30 min at 300 °C to incorporate nitrogen into the surface.

Quaternized SiC (QSiC) samples were prepared by immersing the ammonia plasma-treated samples into a solution containing acetonitrile (25 mL) and allyl bromide (100 µL) for 1 h to quaternize the surface nitrogen. After removing the samples from the quaternization solution, excess solvent and reagent were removed using rinses of isopropanol and deionized water (Fisher Scientific, Pittsburgh, PA, USA).

To coat samples with 50 nm of titanium nitride (TiN), a Kurt J. Lesker CMS 18 (Jefferson Hills, PA, USA) sputter system was used. An allowed TiN target was used in pure Ar ambient at room temperature. A platen bias of 30 V was applied to during TiN deposition to promote film regularity.

Quaternization of the surface nitrogen was performed using the same procedure as for quaternized SiC substrates to yield a quaternized TiN surface. No plasma treatment was required for TiN samples since the surface already contained nitrogen that could be quaternized.

### 2.3. Experimental Design

Five experimental groups were included in this study, these were the non-coated titanium disks as the reference group, titanium nitride coated titanium disks (TiN), quarternized titanium nitride coated titanium disks (QTiN), silicon carbide coated titanium disks (SiC), and quarternized silicon carbide coated titanium disks (QSiC).

### 2.4. Bacterial Corrosion

To study the bacterial corrosion of the coated and non-coated titanium disks, monomicrobial reference strains (ATCC—American Type Culture Collection) of *Porphyromonas gingivalis* (ATCC 33277), *Streptococcus salivarius* (ATCC BAA-1024), *Streptococcus sanguinis* (ATCC BAA-1455), and *Streptococcus mutans* (ATCC 35688) were used. The strains were grown onto Falcon tubes with brain heart infusion broth (BHI—Himedia) for 24 and 48 h at 37 °C. After the growth, each microbial suspension was centrifuged at 4700 rpm for 10 min (MPW-350) to separate the supernatant and microbial suspension. The centrifuge process was performed twice to minimize the quantity of debris. After separation, the microbial suspension adjusted to 10^7^ CFU/mL.

Coated and non-coated titanium disks of each group were sterilized in autoclave (121 °C, 60 min), and each one was distributed on a sterile 12-well plate. For all samples, 1 mL of polymicrobial suspension was added to each well containing a coated or non-coated Ti disk and cultivated for 30 days.

### 2.5. ICP-MS Test

The media was collected every 2 days for 30 days of culturing, and the ICP-MS (Inductively Coupled Plasma Mass Spectrometry) test was performed to find the amount of titanium in the supernatant.

Samples were vortexed immediately prior to aliquoting into 4 mL duplicates. Accurate weights of the aliquots were recorded, since their specific gravity is approximately equal to that of water. One aliquot of each sample was fortified with 50 ng of Ti to verify accurate analysis. One mL of 70% Optima nitric acid (Fisher Scientific, Hampton, NH, USA) and 0.5 mL 30% certified ACS grade hydrogen peroxide (Fisher Scientific) was added to the samples and heated to 100 °C on a dust-free hotplate for 1.5 h. After digestion and cooling, the samples were brought to 15 mL by weight with ultrapure water, again accurately recording dilution weight. Procedural blanks were also tested alongside samples to ensure purity of the reagents. Testing for Ti was accomplished using an Agilent 7900 ICP-MS (Santa Clara, CA, USA) with in-line internal standard addition and utilizing He gas mode, which reduces polyatomic interferences. The Ti isotope to be monitored is 48, the most abundant of naturally occurring Ti isotopes. A set of Ti calibrators with a range 0–10,000 ppb was also be tested to quantitate the samples’ Ti concentrations.

### 2.6. Weight Measurements

The disks were weighed before and after the 30-day bacteria corrosion experiment. Disks were weighed using an analytical balance (AS 220.R2 Analytical Balance, Radwag, Poland). Each disk was weighed three times, the average values were calculated for the weight for each disk. After bacteria corrosion experiment, the disks were cleaned with Triton X-100 and distilled water and dried before the weighing. Weight measurements were done as a verifying step to see if the 30 days immersion in bacteria led to an excessive loss in sample weight (implant weight).

### 2.7. Scanning Electron Microscopy

Non-coated and coated disks were observed under scanning electron microscopy using a MAICE system (JEOL JSM-6400 Scanning Electron Microscope, JEOL LTD, Tokyo, Japan) to identify surface roughness before and after cultivation with a biofilm. Coated (TiN, QTiN, SiC, QSiC) and non-coated disks were incubated for 30 days. The polymicrobial biofilm adhered to the samples was removed by placing the disks inside a Falcon tube with 2 mL of Triton X-100 and vortex for 2 min. The disks were then washed 3× with 2 mL of distilled water. The SEM was operated at 5 kV, spot 3 to 6 (JEOL JSM-6400) and the images were recorded.

### 2.8. Atomic Force Microscopy (AFM)

Topographies of Ti, TiN, QTiN, SiC, and QSiC disks were verified using atomic force microscopy (Bruker/Veeco/Digital Instruments NanoScope V). The AFM was operated in tapping mode using a silicon AFM probe (RTESP-300, Bruker, Billerica, MA, USA), with a radius of less than 10 nm and resonance frequency among 200 and 400 kHz.

The quantitative data were shown as the means ± standard deviations. The statistical differences were calculated using Kruskal–Wallis and Wilcox Rank Sum tests. A *p*-value of ≤0.05 was considered statistically significant.

## 3. Results

### 3.1. Scanning Electron Microscopy

Figure 1 demonstrates the change in surface roughness of Ti reference and Ti coated disks before and after bacterial inoculation. The reference disk (Figure 1F) showed pitting and surface breakdown while these were not evident with the coated samples (Figure 1G–J).

### 3.2. ICP-MS Test

Uncoated Ti disks in contact with bacteria for 30 days released high concentrations of Ti (34.46 ppb) compared with all groups (*p* = 0.003). Ti disks coated with SiC released the lowest amount of Ti with 0.54 ppb for SiC and 3.6 ppb for QSiC (*p* = 0.0009) (Figure 2). TiN (9.57 ppb) and QTiN (11.02 ppb) coated disks presented similar Ti release values (*p* = 0.686).

### 3.3. Weight Measurements

The initial and final weights of coated and non-coated samples were similar in all samples (*p* = 0.348). We could not observe an obvious difference between the initial and final weight, showing no high degradation on Ti disks after 30 days of cultivation with biofilm (Figure 3).

### 3.4. AFM

The roughness of non-coated and coated disks was measured before and after 30 days in contact with a biofilm (Figure 4). Non-coated Ti disks presented similar image Rq than QTiN on Day 0, however the Rq for the non-coated Ti was higher on Day 30.

After 30 days in cultivation with a biofilm, we observed that the image Rq values were higher only for non-coated Ti disks (Figure 4). All coated disks presented image Rq values similar initially (Day 0) and after 30 days of cultivation (Figure 5). The highest Rq was for QTiN with 51.2 mm for day 0 and 54.4 mm for QTiN for day 30.

The spatial parameters were reported in the present study (Table 1), the autocorrelation length (Sal) and the texture aspect ratio (Str). The autocorrelation length was the fastest decay in the perpendicular direction to the texture and the texture aspect ratio was represented as a ratio of the fastest and slowest decay. The non-coated disks (reference) and coated disks seemed to have an anisotropic texture in general (TiN seemed to be a ransom structure, the Str is above 0.5).

The SEM images showed that the non-coated (reference) disk is rough, and the coating was able to planarize the disks conformally. The image roughness demonstrated less roughness (image Rq) from the coating compared with the non-coated sample. As the developed interfacial area ratio (Sdr) showed a similar trend. Usually, the rough surface showed more than 1% Sdr. The Sdr was 4.44% and 4.25% for non-coated and QTiN coated disk, while the Sdr decreased for the other coatings. After bacterial incubation, the non-coated showed higher surface roughness (image Rq) and the higher Sdr, whereas the rest of the samples showed similar results.

## 4. Discussion

Bacteria corrosion on implants is due to biofilm formation, under which the interfacial electrochemical processes start and are followed by microbially-influenced corrosion on substratum metals and alloys [19,20]. Corrosion is a complex process mediated by interaction occurring between anaerobic and aerobic bacteria [7,21]. Underneath the biofilm, the metal surface area where oxygen is limited increases differential aeration cells [22]. The lack of aeration in these areas helps create an anode where corrosion can occur by releasing metal ions into the saliva. Moreover, by combining the released metal ions with the saliva’s chloride ions or the bacteria’s end-products, more reactive species will be generated, and the metal will be corroded further [20,23].

In this study, we verified that a biofilm of *P. gingivalis*, *S. mutans*, *S. salivarius,* and *S. sanguinis* can cause corrosion on the surface of titanium. The influence of sulfides produced by oral microorganisms on the corrosion resistance of pure titanium has been investigated by Harada et al. [4]. After 14 days, XPS analysis exhibited no progression of oxidation on the titanium surface. Levels of titanium were detected in all media; however, no significant differences were verified for all test media and immersion times. The sulfides produced by *P. gingivalis* did not cause titanium corrosion for 14 days. De la Garza-Ramos [24] showed a greater corrosion rate in Ti6Al4V alloy at 4 days of exposure by the medium that contains the *Streptococcus gordonii*.

In this work, a highly sensitive ICP-MS test was performed to check the concentration of Ti in the solution after 30 day cultivation with *P. gingivalis*, *S. mutans*, *S. salivarius*, and *S. sanguinis*. All coated samples presented a lower Ti ppb value compared to the reference samples. We can infer that these coatings may have the potential to protect the Ti surface from the corrosion process. Higher levels of Ti by ICP-MS in biofilm were found in peri-implantitis (48.4 µg/mL) compared to healthy patients (23.8 µg/mL) [21,22].

The application of titanium as an endosseous implant material has been long established. Even though titanium alloys exhibit corrosion resistant properties, due to the stable TiO_2_ layer formed on the surface, they are not immune to corrosive attack [25]. The mechanical stress from chewing and the saliva’s chemical environment can wear down or dissolve the oxide layer over time. Once the TiO_2_ layer is broken down, this layer will not regenerate on the surface, rendering titanium as prone to corrosion as many other base metals [23,26]. To further improve the material’s longevity, thus preventing peri-implantitis, one of the significant strategies focuses on minimizing bacterial adhesion on the dental implant material. Although complete prevention of bacterial adhesion on biomaterials is complicated, various efforts, such as fabricating thin anti-bacterial films or coatings, have been developed to effectively minimize and control the bacterial adhesion biomaterial surface [27,28]. Camargo et al. [29] demonstrated that coatings based on SiC and TiN applied on Ti surfaces were effective against *P. gingivalis* in vitro.

The idea of coating surfaces might be a prospective possibility to decrease the surface roughness of metal alloys and improve their usage for implant prosthetics and other biomedical implants. TiN has been used as a coating for prostheses. Numerous research have reported that TiN has high hardness and remarkable resistance to fatigue and corrosion while exhibiting an intrinsic biocompatibility [30]. SiC coatings for dental application were also reported to exhibit a good corrosion resistance [6,31]. Hsu et al. [32] established that the ceramic samples coated with SiC were not as corroded as the non-coated, suggesting the protective quality of these coatings. Furthermore, SiC on ceramic samples presented antimicrobial activity and biocompatibility [33]. Moreover, the SiC coating has a good resistance to harsh mechanical and chemical environments and is expected to possess great biocompatibility [34,35].

According to the SEM observations in the present study, the surface of non-coated disks presented irregularities, while the coatings on titanium presented smoother surfaces. These findings are corroborated with the results of AFM, which showed smooth coating surfaces, though a homogeneous topography. Regarding roughness, image Rq values revealed SiC was the smoothest surface compared to all the other groups. Furthermore, surface analysis by AFM and SEM showed pitting signs in the non-coated disks. The coatings may have decreased the surface roughness by covering and filling remaining scratches and irregularities of the surface. De la Garza-Ramos [24] indicated that morphology does not present localized corrosion in the Ti6Al4V alloy by SEM observations in contact with *Fusobacterium nucleatum* and *Streptococcus gordonii*.

The findings of this study establish promising results of PECVD thin film coatings, which might be useful in avoiding bacteria corrosion on the Ti surface. However, the results of this study remain limited, and additional studies of corrosion are necessary, in vivo, to determine their influence on clinical outcomes.

## 5. Conclusions

Despite the limitations of this study, PECVD thin film coatings applied to titanium have the potential to minimize the corrosion on implant surfaces. These coatings can be further developed to decrease the prevalence of peri-implantitis disease in dental implants.

## Figures and Tables

**Figure 1 materials-14-00342-f001:**
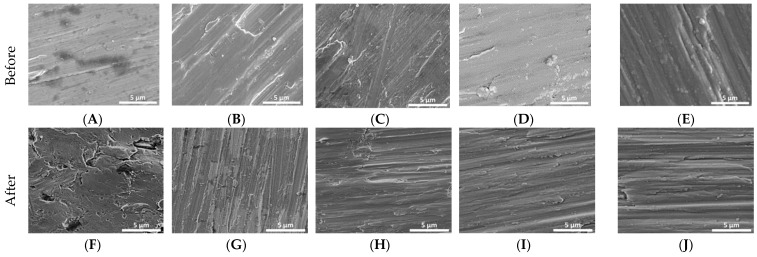
Initial SEM images of Ti disks non-coated and coated (**A**–**E**). SEM images of Ti disks non-coated and coated after 30 days in cultivation of *P. gingivalis*, *S. mutans*, *S. salivarius*, and *S. sanguinis* (**F**–**J**). (**A**,**F**)—non-coated; (**B**,**G**)—coated with titanium nitride (TiN); (**C**,**H**)—coated with quarternized titanium nitride coated titanium (QTiN), (**D**,**I**)—coated with silicon carbide (SiC), and (**E**,**J**)—coated with quaternized SiC (QSiC).

**Figure 2 materials-14-00342-f002:**
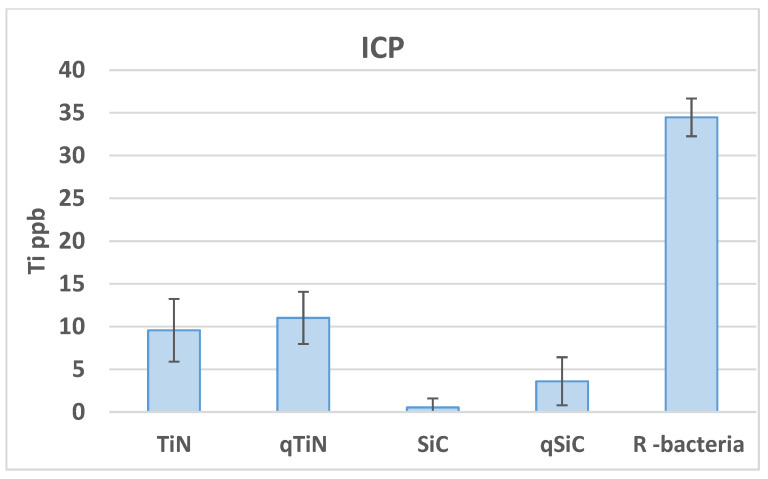
Ti ppb mean values and standard deviation in Ti disks non-coated (R bacteria) and coated (TiN, QTiN, SiC, and QSiC) samples.

**Figure 3 materials-14-00342-f003:**
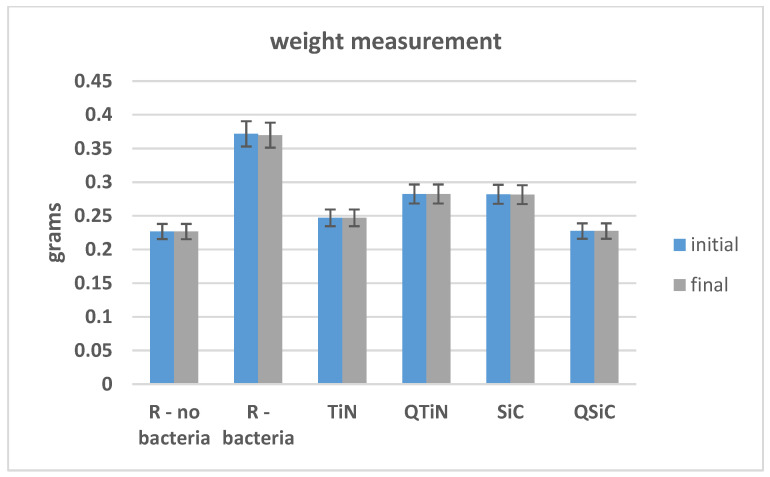
Initial and final weight of Ti disks non-coated and coated before and after 30 days in contact with microorganisms.

**Figure 4 materials-14-00342-f004:**
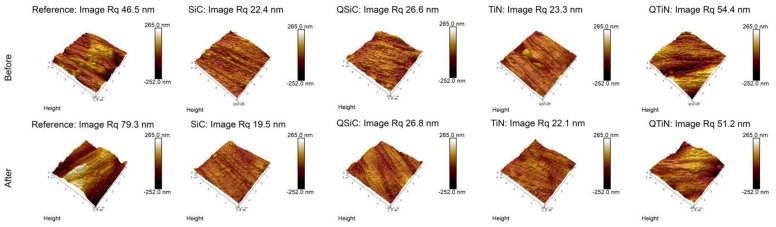
Roughness means values and standard deviation in non-coated (Reference) and coated (TiN, QTiN, SiC, and QSiC) disks before and after 30 days in cultivation with bacteria. AFM images of surface topography of Ti disks non-coated and coated, with amplifications of 10 µm.

**Figure 5 materials-14-00342-f005:**
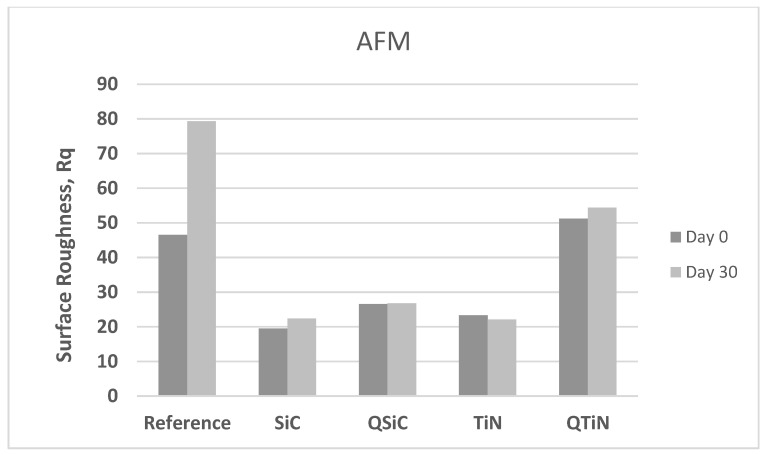
Roughness means values (image Rq) in non-coated (Reference) and coated (TiN, QTiN, SiC, and QSiC) disks initially (Day 0) and after 30 days in cultivation with bacteria.

**Table 1 materials-14-00342-t001:** The spatial parameters of AFM for non-coated (Reference) and coated (TiN, QTiN, SiC, and QSiC) disks initially (Day 0) and after 30 days in cultivation with bacteria.

	Height	Spatial		Hybrid
No Bacteria (Day 0)	Sq (= Image Rq)	Sal (µm)	Str	Sdr (%)
Reference	46.5	0.676	0.297	4.44
SiC	22.4	0.285	0.272	2.41
QSiC	26.6	0.428	0.463	1.95
TiN	23.3	0.535	0.616	1.73
QTiN	54.4	0.549	0.203	4.25
	**Height**	**Spatial**		**Hybrid**
**Bacteria (Day 30)**	**Sq (= Image Rq)**	**Sal (µm)**	**Str**	**Sdr (%)**
Reference	79.3	0.765	0.207	6.52
SiC	19.5	0.485	0.371	2.89
QSiC	26.8	0.511	0.157	2.77
TiN	22.1	0.570	0.781	3.34
QTiN	51.2	0.754	0.363	2.7

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
