# Peer review of "Novel Coatings to Minimize Corrosion of Titanium in Oral Biofilm"

_materials, 2021, doi:10.3390/ma14020342_

Round 1

Reviewer 1 Report

In the reviewed paper, the authors present the results of their studies on a very interesting issue, which is the determination of the influence of biofilm from many microorganisms on the corrosive behaviour of titanium dental implants. Titanium corrosion was evaluated through surface observation changes of titanium substrates and substrates of modified surface using scanning electron microscope (SEM) and atomic force microscopy (AFM). Authors conclude that the thin film coatings have the potential to prevent corrosion on implant surfaces, which is in line with the results of previous studies.

The paper requires the following amendments:

General remark:
- names of bacterial strains should be written in Italics,
- the meaning of the abbreviations and acronyms, which are used in manuscript should firstly clarified, e.g. acronim "QTiN" in Abstract.

Introdution:
- in my opinion, the methods used in studies of corrosion processes, which take place on the surface of titanium implants should be presented and briefly discussed in introduction. This would justify the correctness of the research methods used by the authors,
- What is the novelty of the research carried out?

Materialst and Methods:
- no information about the manufacturer of titanium rods.
- how has roughness been studied?

Results:
- the SEM images analysis results are not discussed,
- what is the goal of weight measurements,.

In my opinion, the results of investigations presented by the authors do not bring anything new to the research carried out so far in this topic.

The reviewed paper can be recommended to publication in Materials only after major revision.

Reviewer 2 Report

Point 01

How many discs were there actually? First it is mentioned 24 discs, then it is mentioned that 5 groups were formed from a total of 90 samples.

Point 02

Why was Rq chosen for the topography analysis? Why only this one? And why not a tridimensional parameter instead of a bi-dimensional one?

Point 03

Why was ANOVA chosen as the statistical method of analysis?

Point 04

The 3 first paragraphs and the fifth paragraph of the Discussion consist of some literature review on the subject without any actual discussion of the findings of the study. The fourth paragraph of the Discussion is simply a repetition of the Results without any actual discussion of the findings of the study. All in all, this Discussion section has no discussion at all.

Point 05

“These coatings can be further developed to decrease the prevalence of peri-implantitis disease in dental implants.”

How can these coatings be further developed? And from which results were the authors able to conclude that these coatings can decrease the prevalence of peri-implantitis disease in dental implants?

Point 06

“PECVD thin film coatings applied to titanium have 194 the potential to minimize the corrosion on implant surfaces”

How much is this clinically significant?

Reviewer 3 Report

Generally speaking the paper is suitable for publication in journal Materials having important future applications in promoting  coatings  (TiN, SIC ) to decrease the prevalence of peri-implantitis disease in dental  Ti implants. The manuscript is well organized and clearly written as well, and such merits are strong points.                             Before publication is a need for revision  taking into account  :

a) to enhance methodology with  Corrosion tests in an environment able to simulate Ti implants medium  of exploitation. It will be in the benefit of the paper quality to introduce electrochemical procedures specific for corrosion such as  open potential detemination, potentiodynamic measurements and electrochemical impedance spectroscopy. Electrochemical parameters characteristic for above techniques introduce  quantified data in tables. In the present manuscript there are only figures.                                                     b) to improved chapter References which has not papers from the last two years despite the fact that in this period of time behavior of coated Ti Implants with TiN (including corrosion) was investigated.

Round 2

Reviewer 1 Report

In revised version of the reviewed manuscript authors took most of my comments into account. In my opinion this paper may be accept to publication in Materials in present form.

Author Response

You can find the answers in the attached file. 

Reviewer 2 Report

Point 01

“Sometimes the use of Sq and Rq can be interchanged.”

No, it cannot.

The randomly distributed structural elements on a surface cannot be seen in 2D. If the application requires a better understanding of the surface structure and a single profile information is not sufficient, 3D measurement should be used (https://www.qualitymag.com/articles/95593-how-to-make-the-right-choice-between-2d-vs-3d-in-surface-metrology). Moreover, 3D irregular rough surfaces produce higher effects than those observed over 2D (https://doi.org/10.1016/j.ijheatfluidflow.2012.04.003). 

Point 02

“Rq is more representative of surface roughness (peaks and valleys) based on the mathematical formula.”

I do not agree. For the analysis of surfaces, at least one of each height, spatial, and hybrid parameter should be presented. Please consult the literature (Wennerberg and Albrektsson).

Point 03

If there were 4 discs per group (amount considered to be insufficient to check normality), ANOVA, a parametric test, should not be used. A non-parametric test needs to be applied instead.

Point 04

The authors had the chance to make adequate changes but instead tried to talk the reviewer into accepting the manuscript without consulting a statistician and without consulting the literature about the appropriate methods to be used.

Author Response

(The authors gave the same response as above.)
